# RT-Sketch: Goal-Conditioned Imitation Learning from Hand-Drawn Sketches

**Priya Sundaresan**[1,3], **Quan Vuong**[2], **Jiayuan Gu**[2], **Peng Xu**[2], **Ted Xiao**[2], **Sean Kirmani**[2], **Tianhe Yu**[2], **Michael Stark**[3], **Ajinkya Jain**[3], **Karol Hausman**[2], **Dorsa Sadigh**[*2], **Jeannette Bohg**[*1], **Stefan Schaal**[*3]

[1]Stanford University, [2]Google DeepMind, [3][Google] Intrinsic
[*]Equal advising, alphabetical order

**Abstract:** Natural language and images are commonly used as goal representations in goal-conditioned imitation learning. However, language can be ambiguous and images can be over-specified. In this work, we study hand-drawn sketches as a modality for goal specification. Sketches can be easy to provide on the fly like language, but like images they can also help a downstream policy to be spatially-aware. By virtue of being minimal, sketches can further help disambiguate task-relevant from irrelevant objects. We present RT-Sketch, a goal-conditioned policy for manipulation that takes a hand-drawn sketch of the desired scene as input, and outputs actions. We train RT-Sketch on a dataset of trajectories paired with synthetically generated goal sketches. We evaluate this approach on six manipulation skills involving tabletop object rearrangements on an articulated countertop. Experimentally we find that RT-Sketch performs comparably to image or language-conditioned agents in straightforward settings, while achieving greater robustness when language goals are ambiguous or visual distractors are present. Additionally, we show that RT-Sketch handles sketches with varied levels of specificity, ranging from minimal line drawings to detailed, colored drawings. For supplementary material and videos, please visit http://rt-sketch.github.io.

**Keywords:** Visual Imitation Learning, Goal-Conditioned Manipulation

## 1 Introduction

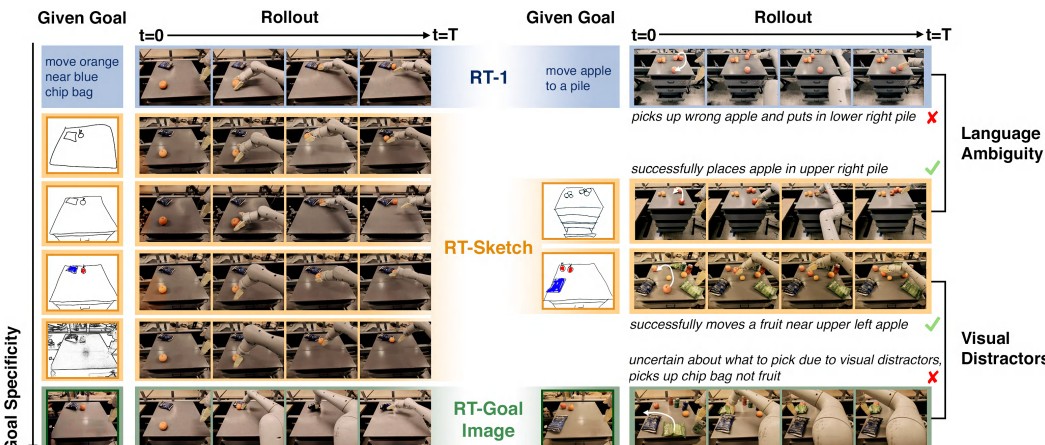

Figure 1: Rollouts showing RT-Sketch's robustness to sketch detail, ambiguous language, and visual distractors.

Robots operating alongside humans in the home or workplace have an immense potential for assistance and autonomy, but careful consideration is needed of what goal representations are easiest *for humans* to convey to robots, and *for robots* to interpret and act upon.

---

Correspondence to `priyasun@stanford.edu`

8th Conference on Robot Learning (CoRL 2024), Munich, Germany.

Instruction-following robots attempt to address this problem using the intuitive interface of natural language commands as inputs to language-conditioned imitation learning policies [1, 2, 3, 4, 5]. For instance, imagine asking a household robot to set the dinner table. A language description such as *"put the utensils, the napkin, and the plate on the table"* is under-specified or ambiguous. It is unclear how exactly the utensils should be positioned relative to the plate or the napkin, or whether their distances to each other matter or not. To achieve this higher level of precision, a user may need to give lengthier descriptions such as *"put the fork 2cm to the right of the plate, and 5cm to the leftmost edge of the table."*, or even online corrections (*"no, you moved too far to the right, move back a bit!"*) [6, 5]. While intuitive, the qualitative nature and ambiguity of language can make it both inconvenient for humans to provide without lengthy instructions or corrections, and for robot policies to interpret for downstream precise manipulation.

Using a goal image (i.e. an image of the scene in its final desired state) to specify objectives and train goal-conditioned imitation learning policies has shown to be quite successful in recent years, with or without language [7, 8, 9]. However, this has its own shortcomings: access to a goal image is a strong prior assumption, and a pre-recorded goal image is tied to a particular environment, making it difficult to reuse for generalization. To summarize: while natural language is highly flexible, it can also be highly ambiguous or require lengthy descriptions. This quickly becomes difficult in long-horizon tasks or those requiring spatial awareness. Meanwhile, goal images over-specify goals in unnecessary detail, leading to the need for internet-scale data for generalization.

To address these challenges, we study *hand-drawn sketches* as a convenient yet expressive modality for goal specification. By virtue of being minimal, sketches are still easy to provide on the fly like language, but allow for more spatially-aware task specification. Like goal images, sketches readily integrate with off-the-shelf policy architectures that take visual input, but provide an added level of goal abstraction that ignores unnecessary pixel-level details. Finally, sketches can inform a policy of task relevant/irrelevant objects based on whether details are included/excluded in a sketch.

In this work, we present RT-Sketch, a goal-conditioned policy for manipulation that takes a hand-drawn sketch of the desired scene as input, and outputs actions. The novel architecture of RT-Sketch modifies the original RT-1 language-to-action Transformer architecture [1] to consume visual goals rather than language, allowing for flexible conditioning on sketches, images, or any other visually representable goals. To enable this, we concatenate a goal sketch and history of observations as input before tokenization, omitting language. We train RT-Sketch on a dataset of 80K trajectories paired with synthetic goal sketches, generated by an image-to-sketch stylization network trained from a few hundred image-sketch pairs.

We evaluate RT-Sketch on six real-world tabletop manipulation tasks subject to a wide range of scene variations. These skills include rearranging objects, placing cans and bottles sideways or upright, and opening and closing drawers. Experimentally, we find that RT-Sketch performs on a similar level to image or language-conditioned agents in straightforward settings. When language instructions are ambiguous, or in the presence of visual distractors (Figure 1, right), we find that RT-Sketch achieves 2.71X and 1.63X higher spatial alignment scores over language or goal image-conditioned policies, respectively (see Fig. 3 (H3/4)). Additionally, we show that RT-Sketch can handle different levels of input specificity, ranging from rough sketches to more scene-preserving, colored drawings (Fig. 1, left). Finally, we also include results suggesting the compatibility of sketches with language, showing promise of multimodal goal specification in the future.

## 2 Related Work

In this section, we discuss prior methods for goal-conditioned imitation learning (IL) and recent efforts towards image-sketch translation, which we build on towards sketch-condition IL.

**Goal-Conditioned Imitation Learning** Reinforcement learning (RL) is not easily applicable in our scenario, as it is nontrivial to define a reward objective which accurately quantifies alignment between a provided scene sketch and states achieved by an agent. We instead focus on IL techniques, particularly goal-conditioned IL [10] which has proven useful in settings where an agent needs to handle different variations of the same task [11]. Examples include moving objects into

different arrangements [1, 2, 5, 12, 9], kitting [13], folding of deformable objects into different configurations [14], and search for different target objects in clutter [15]. However, these approaches tend to condition on either language [1, 4, 5, 3, 16], or image [15] goals. Follow-up work enabled multimodal conditioning on either goal images and language [8], in-prompt images [7], or image embeddings [12, 13, 14]. All of these representations are ultimately derived from raw images or language, which overlooks the potential for more abstract goals like sketches.

Beyond inflexible goal representations, goal-conditioned IL tends to overfit to demonstration data and fails to handle even slight distribution shifts [17]. Ambiguous phrasing or references to unseen objects can significantly degrade the performance of language-conditioned policies, for instance [8, 1]. Goal-image conditioned agents are similarly vulnerable to visual changes like lighting or novel objects and backgrounds [18, 19]. Sketches, on the other hand, are minimal enough to combat visual distractors, yet expressive enough to provide unambiguous goals. [20] explores a multimodal policy conditioned on embeddings of language, images, and sketches obtained from pre-trained models like CLIP [21]. However, these models are not trained on abstract representations like sketches, limiting the policy's performance when conditioned on sketches. Other recent works propose goal-conditioning on *motion-centric* sketches which can either represent the intended direction of positional [22, 23, 24] or joint-level [25, 26] robot movement. We instead consider *scene-centric* sketches, representing the desired visual goal state rather than the desired actions.

**Image-Sketch Conversion**   Sketches have been studied within the computer vision community for object detection [27, 28, 29], visual question answering [30, 31], and scene understanding [32], either in isolation or in addition to text and images. When considering how best to incorporate sketches in IL, an important design choice is whether to take sketches into account (1) at test time (by converting a sketch to another modality compatible with a pre-trained policy), or (2) at train time (by explicitly training a policy conditioned on sketches). For (1), one could first convert a given sketch to a goal image, and then roll out a vanilla goal-image conditioned policy. Existing frameworks tackle sketch-to-image conversion through diffusion models [33, 34], GAN-style approaches [35], or text-to-image synthesis [36, 37]. These models can produce photorealistic images but do not guarantee style transfer, making it unlikely for generated images to match the style of agent observations. These approaches are also susceptible to hallucinated artifacts, introducing distribution shifts [33].

Thus, we instead opt for (2), and consider image-to-sketch conversion techniques for hindsight relabeling of demonstrations. Recently, Vinker et al. [38, 39] propose networks for predicting Bezier curve-based sketches of input images, supervised by a CLIP-based alignment metric. While these approaches generate visually compelling sketches, test-time generation takes on the order of minutes, which does not scale to the typical size of robot learning datasets with hundreds to thousands of trajectories. Meanwhile, conditional generative adversarial networks (cGANs) such as Pix2Pix [40] have proven useful for scalable image-to-image translation. Most related to our work is that of Li et al. [41], which trains a Pix2Pix model to produce sketches from given images on a large crowd-sourced dataset of $5K$ paired images and line drawings. We build on this work to fine-tune an image-to-sketch model that maps robot observations to sketches, with which to train an IL policy.

## 3   Sketch-Conditioned Imitation Learning

**Problem Statement**   We first formalize the problem of learning a manipulation policy conditioned on a goal *sketch* of the desired scene state and a history of interactions. We denote such a policy by $\pi_{\text{sketch}}(a_t|g, \{o_j\}_{j=1}^t)$, where $a_t$ denotes an action at timestep $t$, $g \in \mathbb{R}^{W \times H \times 3}$ is a given goal sketch with width and height $W$ and $H$, and $o_t \in \mathbb{R}^{W \times H \times 3}$ is an observation at $t$. At inference time, the policy takes a given goal sketch along with a history of $D$ previous RGB image observations, and outputs an action. To train such a policy, we assume access to a dataset $\mathcal{D}_{\text{sketch}} = \{g^n, \{(o_t^n, a_t^n)\}_{t=1}^{T^{(n)}}\}_{n=1}^N$ of $N$ successful demonstrations, where $T^{(n)}$ refers to the length of the $n^{th}$ trajectory in timesteps. Each episode of the dataset consists of a given goal sketch and a corresponding demonstration trajectory, with images recorded at each timestep. Our goal is to thus learn the sketch-conditioned imitation policy $\pi_{\text{sketch}}(a_t|g, \{o_j\}_{j=1}^t)$ trained on $\mathcal{D}_{\text{sketch}}$.

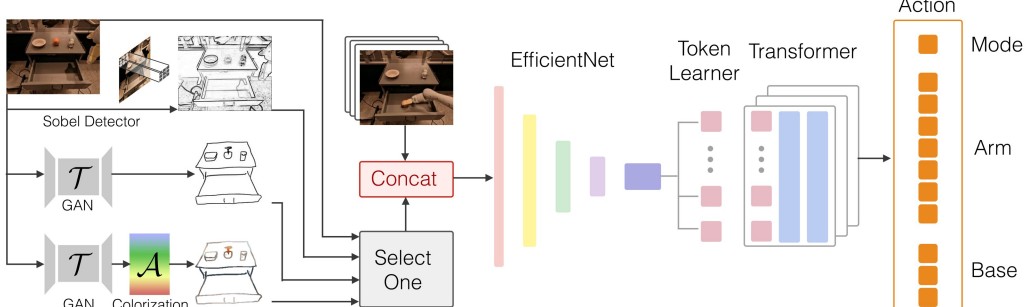

Figure 2: Architecture of RT-Sketch allowing different kinds of visual input. RT-Sketch adopts the Transformer [42] architecture with EfficientNet [43] tokenization at the input, and outputs bucketized actions.

## 3.1 Image-to-Sketch Translation

Training a sketch-conditioned policy requires a dataset of robot trajectories, each paired with a goal sketch. Collecting both demonstration trajectories and manually drawn sketches at scale is impractical. Thus, we instead aim to learn an image-to-sketch translation network $\mathcal{T}(g|o)$ that takes an image observation $o$ and outputs the corresponding goal sketch $g$. This network can be used to post-process an existing dataset of demonstrations $\mathcal{D} = \{\{(o_t^n, a_t^n)\}_{t=1}^{T^{(n)}}\}_{n=1}^N$ with image observations by appending a synthetically generated goal sketch to each demonstration. This produces a dataset for sketch-based IL: $\mathcal{D}_{\text{sketch}} = \{g^n, \{(o_t^n, a_t^n)\}_{t=1}^{T^{(n)}}\}_{n=1}^N$. In practice, we use the existing large-scale dataset of VR-teleoperated robot demonstrations from prior work, which included skills such as object pick and place, placing cans and bottles upright or sideways, and opening and closing cabinets [1]. Prior work previously trained a language-conditioned IL policy RT-1 from this data, but we extend this policy architecture to accommodate sketches, detailed in Section 3.2.

**Assumptions on Sketches**   There are innumerable ways for a human to provide a sketch corresponding to a given image of a scene. For controlled evaluation, we first assume that a given sketch respects the task-relevant contours of an associated image, such that tabletop edges, drawer handles, and task-relevant objects are included and discernible in the sketch. We do not assume contours in the sketch to be edge-aligned or pixel-aligned with those in an image. We do assume that the input sketch consists of black outlines at the very least, with optional color shading. We further assume that sketches do not contain information not present in the associated image, such as hallucinated objects, scribbles, or text, but may omit task-irrelevant details that appear in the original image.

**Sketch Dataset Generation**   To train an image-to-sketch translation network $\mathcal{T}$, we collect a new dataset $\mathcal{D}_{\mathcal{T}} = \{(o_i, g_i^1, \ldots, g_i^{L^{(i)}})\}_{i=1}^M$ consisting of $M$ image observations $o_i$ each paired with a set of goal sketches $g_i^1, \ldots, g_i^{L^{(i)}}$. Those represent $L^{(i)}$ *different* representations of the same image $o_i$, in order to account for the fact that there are multiple, valid ways of sketching the same scene. To collect $\mathcal{D}_{\mathcal{T}}$, we take 500 randomly sampled terminal images from demonstration trajectories in the RT-1 dataset, and manually draw sketches with black lines on a white background capturing the tabletop, drawers, and relevant objects visible on the table. While we personally annotate each robot observation with just one single sketch, we add this data to an existing, much larger non-robotic dataset of paired images and sketches [41]. This dataset captures inter-sketch variation via multiple crowdsourced sketches per image. We do not include the robot arm in our manual sketches, as we find a minimal representation to be most natural. Empirically, we find that our policy can handle such sketches despite actual goal configurations likely having the arm in view. We collect these drawings using a custom digital stylus drawing interface where user draws an edge-aligned sketch over the original image (Appendix Fig. 17) by *tracing outlines*. The final recorded sketch includes the user's strokes in black on a white canvas.

**Image-to-Sketch Training**   We implement the image-to-sketch translation network $\mathcal{T}$ with the Pix2Pix conditional generative adversarial network (cGAN) architecture, which is composed of a

generator $G_{\mathcal{T}}$ and a discriminator $D_{\mathcal{T}}$ [40]. The generator $G_{\mathcal{T}}$ takes an input image $o$, a random noise vector $z$, and outputs a goal sketch $g$. The discriminator $D_{\mathcal{T}}$ is trained to discriminate amongst artificially generated versus ground truth sketches. We utilize the standard cGAN supervision loss to train both [41, 40]: $\mathcal{L}_{\text{cGAN}} = \min_{G_{\mathcal{T}}} \max_{D_{\mathcal{T}}} \mathbb{E}_{o,g}[\log D_{\mathcal{T}}(o, g)] + \mathbb{E}_{o,g}[\log(1 - D_{\mathcal{T}}(o, G_{\mathcal{T}}(o, g))]$. We also add the $\mathcal{L}_1$ loss to encourage the produced sketches to align with ground truth sketches as in [41]. To account for the fact that there may be multiple valid sketches for a given image, we only penalize the minimum $\mathcal{L}_1$ loss incurred across all $L^{(i)}$ sketches provided for a given image as in Li et al. [41]. This is to prevent wrongly penalizing $\mathcal{T}$ for producing a valid sketch that aligns well with one example but not another simply due to stylistic differences in the ground truth sketches. The final objective is a $\lambda$-weighted combination of the average cGAN loss and the minimum alignment loss: $\mathcal{L}_{\mathcal{T}} = \frac{\lambda}{L^{(i)}} \sum_{k=1}^{L^{(i)}} \mathcal{L}_{\text{cGAN}}(o_i, g_i^{(k)}) + \min_{k \in \{1, \dots, L^{(i)}\}} \mathcal{L}_1(o_i, g_i^{(k)})$

In practice, we supplement the 500 manually drawn sketches from $\mathcal{D}_{\mathcal{T}}$ by leveraging the existing larger-scale Contour Drawing Dataset [41]. We refer to this dataset as $\mathcal{D}_{\text{CD}}$, which contains 1000 examples of internet-scraped images containing objects, people, animals from Adobe Stock, paired with $L^{(i)} = 5$ crowd-sourced black and white outline drawings per image collected on Amazon Mechanical Turk (see Appendix Fig. 6 for examples). We first take a pre-trained image-to-sketch translation network $\mathcal{T}_{\text{CD}}$ [41] trained on $\mathcal{D}_{\text{CD}}$, with $L^{(i)} = 5$ sketches per image. Then, we fine-tune $\mathcal{T}_{\text{CD}}$ on $\mathcal{D}_{\mathcal{T}}$, with only $L^{(i)} = 1$ manually drawn sketch per robot observation, to obtain our final image-to-sketch network $\mathcal{T}$. Visualizations of sketches generated by $\mathcal{T}$ are available in Fig. 7.

## 3.2 RT-Sketch

With a way to translate image observations to sketches via $\mathcal{T}$ (Section 3.1), we can automatically augment the RT-1 dataset with goal sketches $\mathcal{D}_{\text{sketch}}$ with which to train our policy RT-Sketch.

**RT-Sketch Dataset** The original RT-1 dataset $\mathcal{D}_{\text{lang}} = \{i^n, \{(o_t^n, a_t^n)\}_{t=1}^{T^{(n)}}\}_{n=1}^{N}$ consists of $N$ episodes with a paired natural language instruction $i$ and demonstration trajectory $\{(o_t^n, a_t^n)\}_{t=1}^{T^n}$. We can automatically hindsight-relabel such a dataset with goal images instead of language goals [44]. Let us denote the last step of a trajectory $n$ as $T^{(n)}$. Then the new dataset with image goals instead of language goals is $\mathcal{D}_{\text{img}} = \{o_{T^{(n)}}^n, \{(o_t^n, a_t^n)\}_{t=1}^{T^{(n)}}\}_{n=1}^{N}$, where we treat the last observation of the trajectory $o_{T^{(n)}}^n$ as the goal $g^n$. To produce a dataset for $\pi_{\text{sketch}}$, we can simply replace $o_{T^{(n)}}^n$ with $\hat{g}^n = \mathcal{T}(o_{T^{(n)}}^n)$ such that $\mathcal{D}_{\text{sketch}} = \{\hat{g}^n, \{(o_t^n, a_t^n)\}_{t=1}^{T^{(n)}}\}_{n=1}^{N}$.

To encourage the policy to afford different levels of input sketch specificity, we in practice produce goals by $\hat{g}^n = \mathcal{A}(o_{T^{(n)}}^n)$, where $\mathcal{A}$ is a randomized augmentation function. $\mathcal{A}$ chooses between simply applying $\mathcal{T}$, $\mathcal{T}$ with colorization during postprocessing (e.g., superimposing a blurred version of the ground truth RGB image over the binary sketch), a Sobel operator [45] for edge detection, or an identity operation, which preserves the original image (Fig. 2). By co-training on all representations, we intend for RT-Sketch to handle a spectrum of specificity going from binary sketches; colorized sketches; edge detected images; and goal images (Appendix Fig. 7).

**RT-Sketch Model Architecture** In our setting, we consider goals provided as sketches rather than language as was done in RT-1. The original RT-1 policy relies on a Transformer architecture backbone [42]. RT-1 first passes a history of $D = 6$ images through an EfficientNet-B3 model [43] producing image embeddings, which are tokenized, and separately extracts textual embeddings and tokens via FiLM [46] and a Token Learner [47]. The tokens are then fed into a Transformer which outputs bucketed actions: a 7-DoF output for the end-effector (x, y, z, roll, pitch, yaw, gripper width), 3-DoF for the mobile base, (x, y, yaw), and 1 mode-switching flag (base movement, arm movement, and termination). To accommodate our change in the input, we omit the FiLM language tokenization altogether. Instead, we concatenate a given visual goal with the history of images as input to EfficientNet, and extract tokens from its output, leaving the rest of the policy architecture unchanged. We train two policies using this architecture (Fig. 2): RT-Sketch refers to our policy trained from sketches, and RT-Goal-Image is a baseline policy trained from goal images.

**Training RT-Sketch** We now train $\pi_{\text{sketch}}$ on $\mathcal{D}_{\pi_{\text{sketch}}}$ from scratch (rather than finetuning an existing backbone) using the same procedure as in RT-1 [1], with the above architectural changes.

We fit the policy using the behavioral cloning objective that minimizes the negative log-likelihood of an action [48]: $J(\pi_{\text{sketch}}) = \sum_{n=1}^{N} \sum_{t=1}^{T^{(n)}} \log \pi_{\text{sketch}}(a_t^n | g^n, \{o_j\}_{j=1}^t)$

# 4 Experiments

We seek to understand the ability of RT-Sketch to perform goal-conditioned manipulation as compared to language or image-conditioned policies. To that end, we test the following four hypotheses:

**H1: RT-Sketch is successful at goal-conditioned IL.** While abstract, we hypothesize that sketches are specific enough to provide manipulation goals to a policy. We thus expect RT-Sketch to perform on a similar level to language (RT-1) or image goals (RT-Goal-Image) in straighforward tasks.

**H2: RT-Sketch is able to handle varying levels of specificity.** Having trained RT-Sketch on sketches of varying levels of specificity, we expect it to be robust against sketch variations for the same scene.

**H3: Sketches enable better robustness to distractors than goal images.** Sketches focus on task-relevant details of a scene, while images capture everything. Therefore, we expect RT-Sketch to provide better robustness than RT-Goal-Image against irrelevant distractors in the environment.

**H4: Sketches are favorable when language is ambiguous.** We expect RT-Sketch to provide a higher success rate compared to ambiguous language inputs when using RT-1.

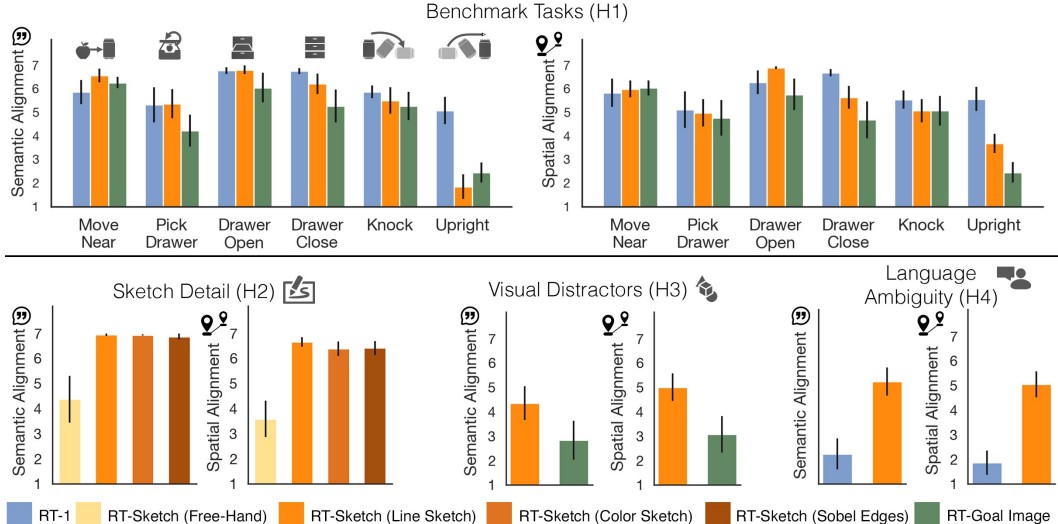

Figure 3: **Goal Alignment Results:** Average Likert scores for different policies rating perceived semantic alignment (**Q1**) and spatial alignment (**Q2**) to a provided goal. Error bars indicate standard error. To back up the visual insights from these barplots, we report additional findings on statistically significant differences between methods from a non-parametric Mann-Whitney U test in Appendix B.

## 4.1 Experimental Setup

**Policies** We compare RT-Sketch to the original language-conditioned agent RT-1 [1], and a goal image-conditioned agent RT-Goal-Image. All policies are trained on a multi-task dataset of $\sim 80$K real-world trajectories manually collected via VR teleoperation using the setup from Brohan et al. [1]. These trajectories span 6 common household object rearrangement tasks: *move X near Y*, *place X upright*, *knock X over*, *open the X drawer*, *close the X drawer*, and *pick X from Y*.

**Evaluation protocol** To fairly compare different policies, we use a shared catalog of heldout evaluation scenarios. Each scenario includes an initial image of the scene, a goal image with objects arranged as desired, a natural language task description, and hand-drawn sketches of the goal. At test time, a human operator retrieves a scenario, aligns the robot and scene using a reference image and a custom visualization utility, and places objects accordingly. We then roll out a policy conditioned on one of the available goals (language, image, sketch, etc.), and record a video for downstream evaluation (see Section 4.2). All experiments utilize the mobile Everyday Robot with an overhead camera and a 7-DoF arm with a parallel jaw gripper. All sketches for evaluation are collected by a single human annotator on a custom drawing interface with a tablet and digital stylus.

**Metrics** Defining a standardized metric for goal alignment is challenging, as binary task success is too coarse, and image-similarity metrics like CLIP [21] can be unreliable. To address this, we

use two metrics for goal alignment. First, we measure the pixel distance between object centroids in achieved versus ground truth goal images (see Fig. 9 in Appendix). Although using object detectors is possible, we avoid this to prevent conflating detection errors (e.g., imprecise or incorrect bounding boxes) with policy errors. Instead, we report results using manually annotated keypoints on achieved and reference goal images. Second, we gather human assessments of perceived goal alignment through two Likert questions [49], rated from 1-7 (Strongly Disagree/Agree).

(**Q1**) *The robot achieves **semantic alignment** with the given goal during the rollout.*

(**Q2**) *The robot achieves **spatial alignment** with the given goal during the rollout.*

| Skill | Spatial Precision (RMSE in px.) | | | Failure Occurrence (Excessive Retrying) | | |
|---|---|---|---|---|---|---|
| | RT-1 | RT-Sketch | RT-Goal-Image | RT-1 | RT-Sketch | RT-Goal-Image |
| Move Near | $5.43 \pm 2.15$ | $\mathbf{3.49 \pm 1.38}$ | $3.89 \pm 1.16$ | **0.00** | 0.06 | 0.33 |
| Pick Drawer | $5.69 \pm 2.90$ | $4.77 \pm 2.78$ | $\mathbf{4.74 \pm 2.01}$ | **0.00** | 0.13 | 0.20 |
| Drawer Open | $4.51 \pm 1.55$ | $\mathbf{3.34 \pm 1.08}$ | $4.98 \pm 1.16$ | **0.00** | **0.00** | 0.07 |
| Drawer Close | $\mathbf{2.69 \pm 0.93}$ | $3.02 \pm 1.35$ | $3.71 \pm 1.67$ | **0.00** | **0.00** | 0.07 |
| Knock | $7.39 \pm 1.77$ | $\mathbf{5.36 \pm 2.74}$ | $5.63 \pm 2.60$ | **0.00** | 0.13 | 0.40 |
| Upright | $7.84 \pm 2.37$ | $5.08 \pm 2.08$ | $\mathbf{4.18 \pm 1.54}$ | 0.06 | **0.00** | 0.27 |
| Visual Distractors | - | $\mathbf{4.78 \pm 2.17}$ | $7.95 \pm 2.86$ | - | **0.13** | 0.67 |
| Language Ambiguity | $8.03 \pm 2.52$ | $\mathbf{4.45 \pm 1.54}$ | - | 0.40 | **0.13** | - |

Table 1: **Spatial Precision / Failure Occurrence:** We report (1) the spatial precision (root mean squared pixel error, RMSE) of the centroids of manipulated objects in achieved vs. given reference goal images (left, darker=more precise) and (2) the occurrence of *excessive retrying* failures (right, bold=least failure-prone).

For **Q1**, we present labelers with the policy rollout video along with the language goal. To answer **Q2**, we present labelers with a policy rollout video side-by-side with a visual goal (ground truth image, sketch, etc.). A policy can for instance achieve high semantic alignment for the language goal *place can upright* as long as the can ends up in the right orientation, but will not achieve spatial alignment unless the can is additionally in the correct position on the table.

Appendix Fig. 18 visualizes the assessment interface. We perform these human assessment surveys across 62 unpaid individuals (non-expert, unfamiliar with our system) who are blind to whether they assess our approach or a baseline. We assign 8-12 people to evaluate each of the 6 different manipulation skills considered. Note that this evaluation is NOT a *user study*, as we are not attempting to study humans, and is merely used as a fair means of *labeling* rollouts to assess goal alignment.

### 4.2 Experimental Results

In this section, we present our findings related to the hypotheses of Section 4 by quantifying precision (Table 1, Table 2) and goal alignment (Fig. 3)) across policies.

**H1**: We evaluate all policies on each of the 6 skills on 15 different evaluation catalog scenarios per skill, varying objects (16 unique in total) and their placements. Overall, RT-Sketch performs comparably to RT-1 and RT-Goal-Image in both semantic (**Q1**) and spatial alignment (**Q2**), achieving average ratings from 'Agree' to 'Strongly Agree' for nearly all skills (Fig. 3 (top)). The exception is *upright*; both RT-Sketch and RT-Goal-Image tend to *position* cans or bottles appropriately, without realizing the need for *reorientation* (Appendix Fig. 10). This results in low semantic alignment but somewhat higher spatial alignment ( Fig. 3 (top), darker gray in Table 1 (left)). RT-1, on the other hand, reorients cans and bottles successfully, but at the expense of higher spatial error (Appendix Fig. 10, light color in Table 1 (left)). With RT-Goal-Image in particular, we also observe the occurrence of *excessive retrying behavior*, in which a policy attempts to align the current scene with a given goal with retrying actions that inadvertently disturb the scene, knocking objects off the table or undoing task progress. In Table 1, we report the proportion of rollouts in which this occurs (via manual inspection) across all policies. RT-Goal-Image is most susceptible, as a result of over-attending to pixel-level details, while RT-Sketch and RT-1 are far less vulnerable, given the higher-level goal abstractions that sketches and language offer.

**H2**: Next, we assess RT-Sketch's ability to handle varying levels of sketch detail. Across 5 trials of the *move near* and *open drawer* skills, we see

| Skill | Free-Hand | Line Sketch | Color Sketch | Sobel Edges |
|---|---|---|---|---|
| Move Near | $7.21 \pm 2.76$ | $3.49 \pm 1.38$ | $3.45 \pm 1.03$ | $\mathbf{3.36 \pm 0.66}$ |
| Drawer Open | $3.75 \pm 1.63$ | $3.34 \pm 1.08$ | $2.48 \pm 0.50$ | $\mathbf{2.13 \pm 0.25}$ |

Table 2: **RT-Sketch Spatial Precision across Sketch Types:** The relatively small differences in policy precision (RMSE) across different sketch types (i.e. minimal line sketches vs. edge-detected images) suggests RT-Sketch's robustness to input specificity (darker=better).

in Table 2 that many different sketch types result in reasonable levels of spatial precision, particularly: free-hand sketches drawn completely free-form on a blank canvas, line sketches drawn by tracing an image, line sketches with color shading, and edge-detected images. Appendix Fig. 17 shows the interface used to sketch, and a detailed breakdown of the differences. As expected, Sobel edge-detected images incur the least error, but they are impractical and merely represent an

upper-bound in terms of sketch detail. Even free-hand sketches, which do not necessarily preserve perspective projection, and line sketches, which are far sparser in detail, are not far behind in terms of precision or alignment ratings. This is reflected in the Likert ratings (Fig. 3 (left, bottom)) of free-hand sketches (around 4 on average), and line sketches (nearly 7 – "Strongly Agree" on average). Adding color to line sketches does not further improve performance, but leads to interesting behavioral differences (see Appendix Fig. 11). In Appendix A.2, we also evaluate RT-Sketch on sketches drawn by 6 different individuals whose sketches were never seen during training and observe little-to-no policy performance drop-off compared to in-distribution sketches.

**H3**: Next, we compare the robustness of RT-Sketch and RT-Goal-Image to the presence of visual distractors. On 15 *move X near Y* trials from the evaluation catalog, we introduce $5 - 9$ distractor objects into the initial visual scene, replicating the setup of the RT-1 generalization experiments referred to as *medium-high* difficulty [1]. In Table 1 (left, bottom), we see that RT-Sketch exhibits far lower spatial errors on average, while producing higher semantic and spatial alignment scores over RT-Goal-Image (Fig. 3 (middle, bottom)). RT-Goal-Image is easily confused by the distribution shift introduced by distractor objects, and often cycles between picking up and putting down the wrong object. RT-Sketch, on the other hand, ignores task-irrelevant objects not captured in a sketch and completes the task in most cases (see Appendix Fig. 12).

**H4**: Finally, we evaluate whether sketches as a representation are favorable when language goals alone are ambiguous. On 15 evaluation catalog scenarios, we consider 3 types of language ambiguity: instance (**T1**) (e.g., *move apple near orange* when multiple orange instances are present), somewhat out-of-distribution (OOD) phrasing (**T2**) (e.g., *move left apple near orange*), and highly OOD phrasing (**T3**) (e.g., *complete the rainbow*) (see Appendix Fig. 13). Directional cues (i.e. 'left') should intuitively help resolve ambiguities, but were unseen during RT-1 training [1], and hence are out-of-distribution. In these scenarios, RT-Sketch achieves nearly half the error of RT-1 (Table 1 (left, bottom)), and a 2.33-fold and 2.71-fold score increase for semantic and spatial alignment, respectively (Fig. 3 (right, bottom)). For **T1** and **T2** scenarios, RT-1 often tries to pick up an instance of any object mentioned in the task string, but fails to make further progress (Appendix Fig. 14). This suggests the utility of sketches to express new, unseen goals with minimal overhead, when language can easily veer out of distribution (Appendix Fig. 15).

**Expanded Evaluation Results** On our website, we include a range of additional results highlighting: (1) multimodal goal specification using both sketches and language, (2) deployment on new robot embodiments, including a Franka Panda robot (3), compatibility with alternative IL backbones, specifically Diffusion Policy [50], (4) performance on new tasks, and (5) support for alternative sketch types like arrows.

### 4.3 Limitations and Failure Modes

Firstly, the image-to-sketch generation network used in this work is fine-tuned on a dataset of sketches provided by a single human annotator. Although we empirically show that RT-Sketch can handle sketches drawn by other annotators (Appendix A.2), we have yet to investigate the effects of training RT-Sketch at scale with sketches drawn by different people. An additional challenge is handling extremely minimal sketches. These kinds of sketches remain difficult for our policy to handle due to obvious perspective changes or missing details. Applying our existing sketch augmentations at more extremes may help further address this class of sketches. Secondly, we note that RT-Sketch shows some inherent biases towards performing certain skills it was trained on, and generalizing to completely unseen or complex tasks remains challenging. However, we posit that addressing these issues may require *policy-level* rather than just *goal-level* improvements. For a detailed breakdown of RT-Sketch's limitations and failure modes, please see Appendix F).

## 5 Conclusion

We propose RT-Sketch, a goal-conditioned policy for manipulation that takes a hand-drawn scene sketch as input, and outputs actions. We do so by developing a scalable way to generate paired sketch-trajectory training data via an image-to-sketch translation network, and modifying the existing RT-1 architecture to take visual information as an input. Empirically, RT-Sketch not only performs comparably to existing language or goal-image conditioning policies for a number of manipulation skills, but is amenable to different degrees of sketch fidelity, and more robust to visual distractors or ambiguities. Our rigorous evaluations comprise 400 cumulative robot rollouts, evaluated across 62 annotators (over 8 cumulative hours). Future work will focus on multimodal goal specification and moving towards even more abstract goal representations, detailed in Appendix C.

## Acknowledgments

We thank Keegan Go, Kaylee Burns, Gautam Salhotra, and Kevin Tracy for the useful suggestions and feedback on this manuscript. Also, we would like to thank Cheri Tran, Emily Perez, Grecia Salazar, Jaspiar Singh, and Jodilyn Peralta for their extensive contributions to evaluations. Priya Sundaresan is supported by an NSF GRFP and was hosted as a PhD Resident at Intrinsic throughout the duration of this project.

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

# A  Additional Evaluations

In this section, we highlight the scale of our evaluations, additional findings from stress-testing RT-Sketch on sketches drawn by different individuals, and results from extending our policy to accommodate sketch+language conditioning.

## A.1  Experiments At A Glance

Cumulatively, our results encompass the following: H1 experiments comprise 270 rollouts (6 skills x 15 trials x 3 methods), H2 comprises 40 rollouts (2 skills x 5 trials x 4 sketch types), H3 comprises 30 rollouts (15 trials x 2 methods), and H4 comprises 30 rollouts (15 trials x 2 methods). All rollouts are cumulatively evaluated across 62 labelers (split across H1-4).

## A.2  Robustness to Input Sketches

To test whether RT-Sketch generalizes to sketches drawn by different individuals, we collect 30 *line sketches* (drawn via tracing) by 6 different annotators (whose sketches were never seen during training) on 5 trials of the *move near* scenario.

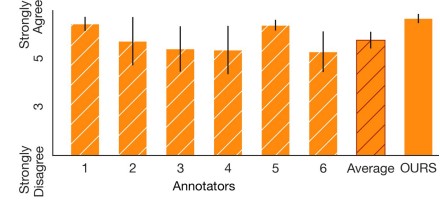

We obtain the resulting rollouts produced by RT-Sketch with these sketches as input. Across ratings, RT-Sketch achieves high spatial alignment on sketches drawn by other annotators. Notably, the performance between sketches drawn by different annotators is similar, as well as the average across annotators compared to original policy performance on our original sketches (Fig. 4).

Figure 4: **Sketches Drawn by Other Annotators**

## A.3  Multimodal Goal Specification: Sketches + Language

We train a sketch-and-language conditioned model by modifying the RT-1 architecture to use FiLM along with EfficientNet layers to tokenize both visual input and language, and concatenate them at the input. In H1 experiments (Fig. 3), we evaluate all policies on the *upright* skill, where the robot must place a can or bottle from a sideways orientation initially to an upright orientation at a desired location on the table. While RT-1 typically can reorient the can/bottle properly, it struggles to place the item in the intended location on the table, as reflected in this policy's spatial imprecision in Table 1. Meanwhile, RT-Sketch struggles to reorient the can/bottle, since an imperfect sketch may fail to specify the exact desired orientation, but often places the can/bottle in the desired location. In Fig. 5, we see that while language alone (i.e. "place the can upright") can be ambiguous in terms of spatial placement, and a sketch alone does not encourage reorientation, we empirically see that the joint policy is better able to address the limitations of either modality alone. A similar pattern emerges for *pick drawer* (Fig. 5).

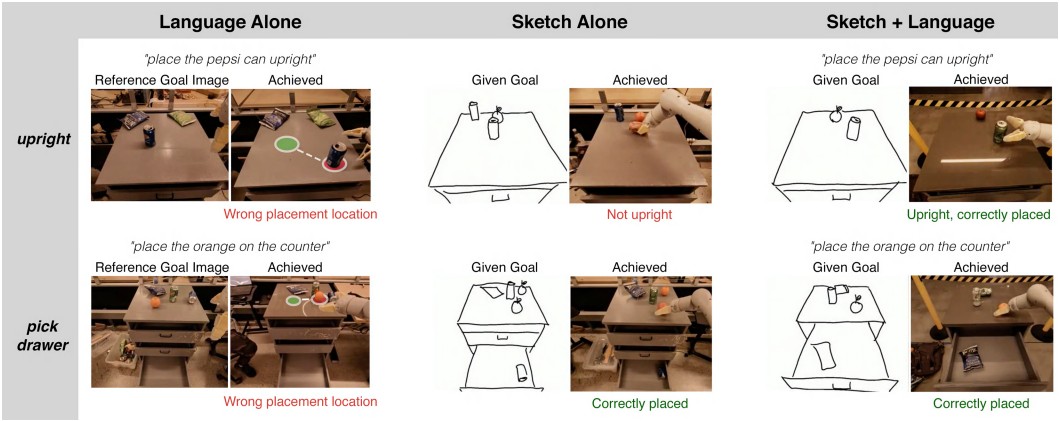

Figure 5: **Multimodal Goal Specification: Sketch+Language**: Empirically, we find that while a language-only policy can struggle with spatial precision, and a sketch-only policy can fail to interpret intended object orientations from a sketch alone, a multimodal policy is better able to address the limitations of both.

# B  Additional Results: Goal Alignment

In addition to the goal alignment results reported in Fig. 3 which are based on average Likert ratings, we additionally conduct a non-parametric Mann-Whitney U (MWU) test with $\alpha = 0.05$ for H1-4

to evaluate the differences in goal alignment ratings across methods. This kind of statistical test is suitable for ordinal data and does not make specific assumptions on the normality or variance of the data distributions.

## B.1 H1 Findings

The H1 experiments aim to evaluate how RT-Sketch compares to RT-1 and RT-Goal-Image on the standard RT-1 tabletop manipulation benchmark [1]. We conduct a MWU test under the null hypothesis that there is no difference in the goal alignment ratings from labelers across the methods. In Appendix Table 3 and Appendix Table 4, we report the pairs of methods for which the ratings yield a p-value of $< 0.05$, rejecting the null hypothesis, along with their $U$-statistic.

Table 3: **H1: RT-1 Benchmark - Semantic Alignment**

| Skill | Method Pair | Stat. | p-value |
|---|---|---|---|
| Move Near | | | |
| Pick Drawer | (RT-1, RT-Goal Img) | 5298.0 | $1.49 \times 10^{-3}$ |
| Drawer Open | (RT-1, RT-Goal Img) | 4797.0 | $1.22 \times 10^{-3}$ |
| Drawer Close | (RT-1, RT-Goal Img) | 4089.5 | $2.01 \times 10^{-8}$ |
| Knock | | | |
| Upright | (RT-1, RT-Sketch) | 16855.0 | $9.49 \times 10^{-29}$ |
| | (RT-1, RT-Goal Img) | 10052.0 | $2.80 \times 10^{-18}$ |
| | (RT-Sketch, RT-Goal Img) | 7210.5 | $5.62 \times 10^{-7}$ |

Table 4: **H1: RT-1 Benchmark - Spatial Alignment**

| Skill | Method Pair | Stat. | p-value |
|---|---|---|---|
| Move Near | | | |
| Pick Drawer | | | |
| Drawer Open | (RT-1, RT-Goal Img) | 4761.5 | $4.59 \times 10^{-3}$ |
| Drawer Close | (RT-1, RT-Sketch) | 7780.0 | $1.82 \times 10^{-5}$ |
| | (RT-1, RT-Goal Img) | 4869.0 | $3.62 \times 10^{-10}$ |
| Knock | | | |
| Upright | (RT-1, RT-Sketch) | 15085.0 | $1.55 \times 10^{-14}$ |
| | (RT-1, RT-Goal Img) | 10656.0 | $1.32 \times 10^{-23}$ |

We conclude that for 5 of 6 and 4 of 6 skills, the null hypothesis is confirmed for semantic and spatial alignment ratings, respectively, suggesting that there is no dropoff in performance with sketches compared to traditional modalities. We do observe that for the *upright* skill, the rating difference between RT-Sketch and RT-1 is significant, and RT-Sketch suffers a slight performance drop as re-orientation is particularly difficult to infer from a sketch alone. However, we have since addressed this challenge with a policy conditioned on both sketches and language, which performs reorientation better than sketches-alone and with more spatial precision than language-alone (Appendix A.3).

The highlighted rows above indicate when the goal alignment ratings for RT-Sketch compared to either RT-1 or RT-Goal-Image were found to be statistically significant. Notably, there are very few such findings, in alignment with H1. This is in accordance with what we observe Fig. 3: nearly no noticeable difference in performance between methods for most of the skills, and the slightly better performance of RT-1 compared to RT-Sketch (and the slightly better performance of RT-Sketch compared to RT-Goal-Image) for the *upright* skill.

Table 5: **H2: Robustness to Sketch Specificity - Semantic Alignment**

| Pair | Stat. | p-value |
|---|---|---|
| Free-Hand, Line Sketch | 1059.0 | $9.58 \times 10^{-12}$ |
| Free-Hand, Colored Sketch | 960.0 | $2.54 \times 10^{-10}$ |
| Free-Hand, Sobel Edges | 1099.5 | $9.16 \times 10^{-11}$ |
| Line Sketch, Colored Sketch | - | - |
| Line Sketch, Sobel Edges | - | - |
| Colored Sketch, Sobel Edges | - | - |

Table 6: **H2: Robustness to Sketch Specificity - Spatial Alignment**

| Pair | Stat. | p-value |
|---|---|---|
| Free-Hand, Line Sketch | 478.0 | $5.18 \times 10^{-17}$ |
| Free-Hand, Colored Sketch | 567.5 | $3.49 \times 10^{-13}$ |
| Free-Hand, Sobel Edges | 629.0 | $3.09 \times 10^{-14}$ |
| Line Sketch, Colored Sketch | - | - |
| Line Sketch, Sobel Edges | - | - |
| Colored Sketch, Sobel Edges | - | - |

## B.2   H2 Findings

For H2 experiments, we evaluate RT-Sketch's robustness to the input specificity of the sketch. We find that across the 4 sketch types, the only pairings which garner statistically significant differences in ratings are free-hand sketches as compared to other types (Appendix Table 5 and Appendix Table 6). This is natural given the drastic perspective and geometric differences of free-hand sketches compared to those which are *traced* or derived from a transform of the goal image itself (edge detection).

However, there are notably no statistically significant pairings between line-sketches and even the most detailed type of input representation we evaluate (Sobel Edges). This suggests that RT-Sketch is indeed able to handle a range of input specificity levels, and more importantly that RT-Sketch can deal with representations that are minimal and imperfect.

Table 7: **H3: Visual Distractors**

| Alignment | Method Pair | Stat. | p-value |
|---|---|---|---|
| Semantic | RT-Sketch, RT-Goal Img. | 20622.5 | $4.62 \times 10^{-8}$ |
| Spatial | RT-Sketch, RT-Goal Img. | 22233.0 | $3.07 \times 10^{-12}$ |

Table 8: **H4: Language Ambiguity**

| Alignment | Method Pair | Stat. | p-value |
|---|---|---|---|
| Semantic | RT-Sketch, RT-1 | 4756.0 | $1.34 \times 10^{-24}$ |
| Spatial | RT-Sketch, RT-1 | 3680.5 | $3.53 \times 10^{-30}$ |

## B.3   H3 and H4 Findings

Finally, we conduct a MWU test over the semantic/spatial goal alignment ratings between RT-Sketch and RT-Goal-Image in the setting of visual distractors (H3, Appendix Table 7) as well as RT-Sketch and RT-1 in the setting of language ambiguity (H4, Appendix Table 8). We hypothesize that RT-Sketch does indeed achieve *higher* ratings than baselines in these settings, as sketches are by nature 1) minimal, which may enable emergent robustness to distractors, and 2) agnostic to language.

We do find a statistically significant difference across semantic and spatial ratings (highlighted in orange), concluding that RT-Sketch is favorable to traditional modalities in these particular settings.

## B.4   Summary of Mann-Whitney U Findings

In short, the additional findings from conducting more thorough MWU testing over H1-4 align very closely with what we observe and report in Fig. 3 and suggest the merits of sketches across a range of scenarios.

## C   Future Directions

Learning a policy conditioned on view-invariant sketches can be an initial step before moving to even more abstract representations like schematics or diagrams for assembly tasks. Additionally, alternative ways to condition on sketches is a powerful avenue for future work. RT-Sketch currently only considers goal observations in sketch space, but projecting all observations to a sketch-based

or latent space is another underexplored but promising direction. Sketches are not without their own limitations, however, as ambiguity due to omitted details or poor quality sketches are persistent challenges. In the future, we are excited to continue exploring multimodal goal specification which can leverage the benefits of language, sketches, and other modalities to jointly resolve ambiguity from any single modality alone. This may include both end-to-end approaches that can jointly condition on multiple modalities, or hierarchical strategies that can leverage the spatial awareness of sketches and the summarization capabilities of VLMs to supplement ambiguous language with more informed descriptions derived from visual observations of a sketch. Lastly, exploring what combination of modalities humans prefer to use when providing goals, and how best they capture intent, is an important future direction not addressed in this work.

## D    Sketch Goal Representations

Since the main bottleneck to training a sketch-to-action policy like RT-Sketch is collecting a dataset of paired trajectories and goal sketches, we first train an image-to-sketch translation network $\mathcal{T}$ mapping image observations $o_i$ to sketch representations $g_i$, discussed in Section 3. To train $\mathcal{T}$, we first take a pre-trained network for sketch-to-image translation [41] trained on the ContourDrawing dataset of paired images and edge-aligned sketches (Fig. 6). This dataset contains $L^{(i)} = 5$ crowd-sourced sketches per image for 1000 images. By pre-training on this dataset, we hope to embed a strong prior in $\mathcal{T}$ and accelerate learning on our much smaller dataset. Next, we finetune $\mathcal{T}$ on a dataset of 500 manually drawn line sketches for RT-1 robot images. We visualize a few examples of our manually sketched goals in Fig. 7 under 'Line Drawings'.

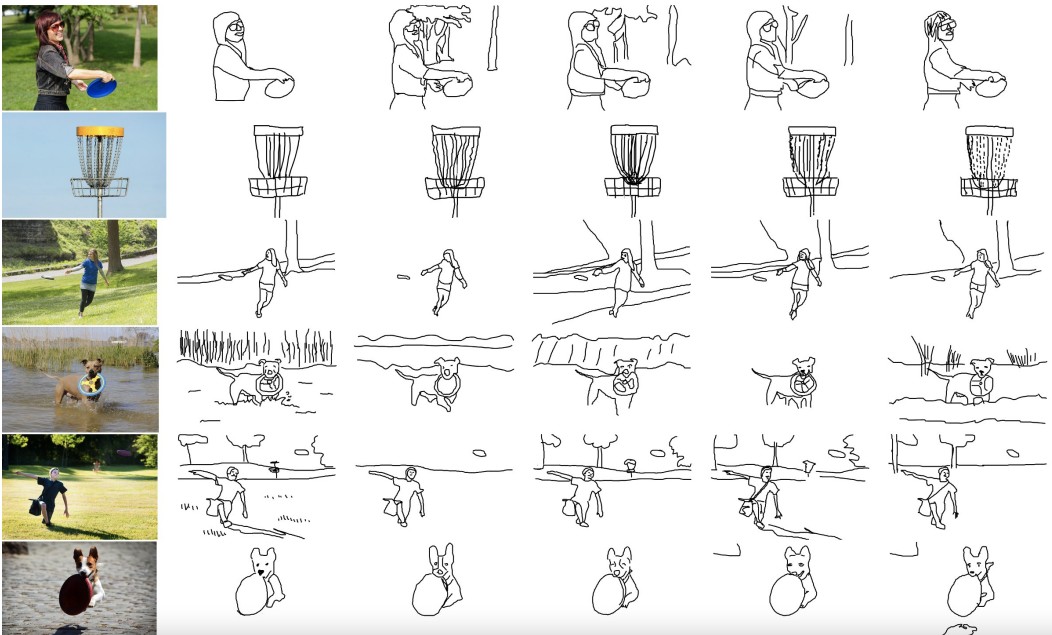

Figure 6: **ContourDrawing Dataset**: We visualize 6 samples from the ContourDrawing Dataset from [41]. For each image, 5 separate annotators provide an edge-aligned sketch of the scene by outlining on top of the original image. As depicted, annotators are encouraged to preserve main contours of the scene, but background details or fine-grained geometric details are often omitted. Li et al. [41] then train an image-to-sketch translation network $\mathcal{T}$ with a loss that encourages aligning with at least one of the given reference sketches.

Notably, while we only train $\mathcal{T}$ to map an image to a black-and-white line sketch $\hat{g}_i$, we consider various augmentations $\mathcal{A}$ on top of generated goals to simulate sketches with varied colors, affine and perspective distortions, and levels of detail. Fig. 7 visualizes a few of these augmentations, such as automatically colorizing black-and-white sketches by superimposing a blurred version of the original RGB image, and treating an edge-detected version of the original image as a generated sketch to simulate sketches with a lot of details. We generate a dataset for training RT-Sketch by 'sketchifying' hind-sight relabeled goal images via $\mathcal{T}$ and $\mathcal{A}$.

Although RT-Sketch is only trained on generated line sketches, colorized line sketches, edge-detected images, and goal images, we find that it is able to handle sketches of even greater diversity.

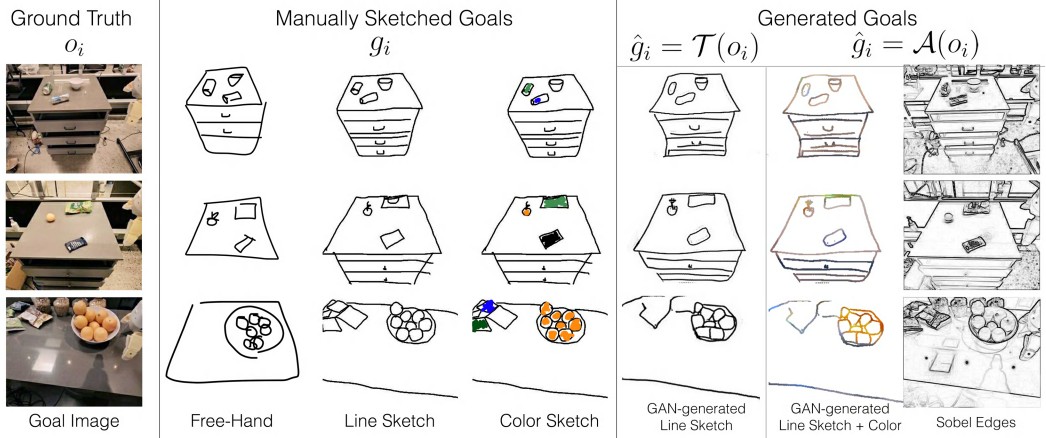

Figure 7: **Visual Goal Diversity**: RT-Sketch is capable of handling a variety of visual goals at both train and test time. RT-Sketch is trained on generated and augmented images like those shown on the right below 'Generated Goals'. But it can also interpret free-hand, line sketches, and colored sketches at test time such as those on the left below 'Manually Sketched Goals'.

This includes non-edge aligned free-hand sketches and sketches with color infills, like those shown in Fig. 7.

### D.1 Alternate Image-to-Sketch Techniques

The choice of image-to-sketch technique we use is critical to the overall success of the RT-Sketch pipeline. We experiment with various other techniques before converging on the above approach.

Recently, two recent works, CLIPAsso [38] and CLIPAScene [39] explore methods for automatically generating a sketch from an image. These works pose sketch generation as inferring the parameters of Bezier curves representing "strokes" in order to produce a generated sketch with maximal CLIP-similarity to a given input image. These methods perform a per-image optimization to generate a plausible sketch, rather than a global batched operation across many images, limiting their scalability. Additionally, they are fundamentally more concerned with producing high-quality, aesthetically pleasing sketches which capture a lot of extraneous details.

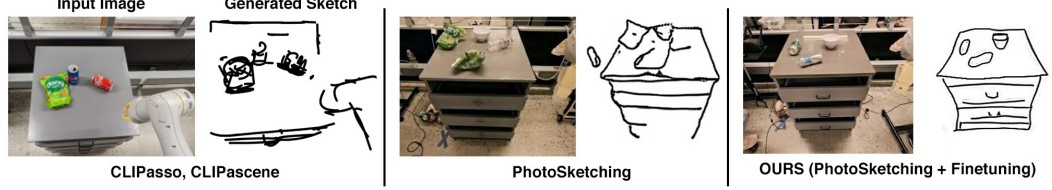

Figure 8: **Alternate Image-to-Sketch Techniques**

We, on the other hand, care about producing a minimal but reasonable-quality sketch. The second technique we explore is trying the pre-trained Photosketching GAN [41] on internet data of paired images and sketches. However, this model output does not capture object details well, likely due to not having been trained on robot observations, and contains irrelevant sketch details. Finally, by finetuning this PhotoSketching GAN on our own data, the outputs are much closer to real, hand-drawn human sketches that capture salient object details as minimally as possible. We visualize these differences in Fig. 8.

## E  Evaluation Visualizations

To further interpret RT-Sketch's performance, we provide visualizations of the precision metrics and experimental rollouts. In Fig. 9, we visualize the degree of alignment RT-Sketch achieves, as quantified by the pixelwise distance of object centroids in achieved vs. given goal images. In Fig. 10, Fig. 11, Fig. 12, and Fig. 14, we visualize each policy's behavior for **H1, H2, H3** and **H4**, respectively. Fig. 13 visualizes the four tiers of difficulty in language ambiguity that we analyze for **H4**.

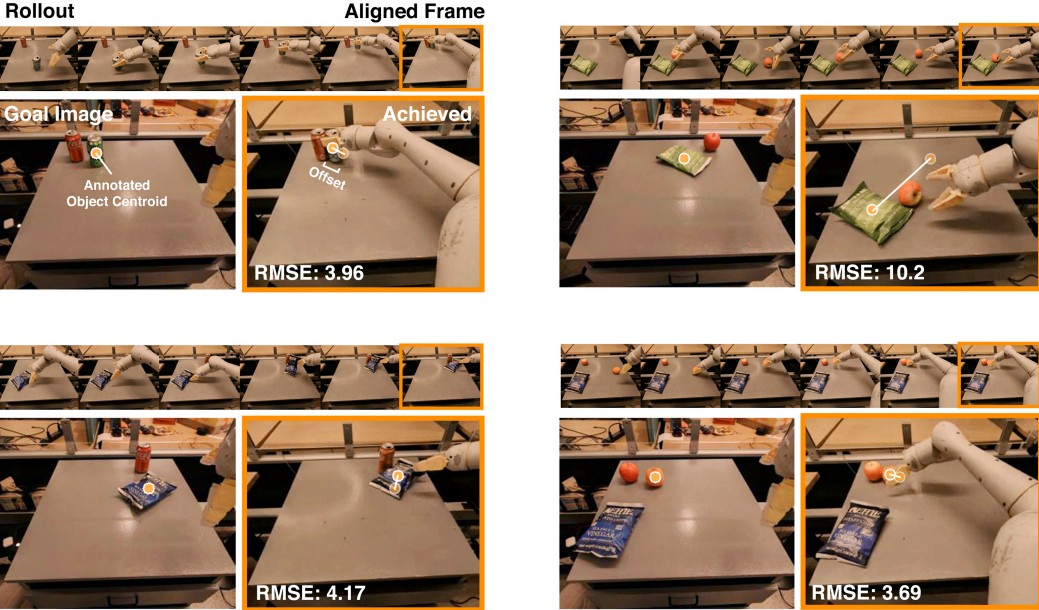

Figure 9: **Spatial Precision Visualization**: We visualize four trials of RT-Sketch on the Move Near skill, along with the measured spatial precision in terms of RMSE. To evaluate spatial precision, we have a human annotator annotate the frame that is visually most aligned, and then keypoints for the object that was moved in this frame and in the provided reference goal image. For each of the four trials, we visualize the rollout frames until alignment is achieved, along with the labeled object centroids and the offset in achieved vs. desired positions. The upper right example shows a failure of RT-Sketch in which the apple is moved instead of the chip bag, incurring a high RMSE. These visualizations are intended to better contextualize the numbers from Table 1.

## F    RT-Sketch Failure Modes and Limitations

While RT-Sketch  is performant at several manipulation benchmark skills, capable of handling different levels of sketch detail, robust to visual distractors, and unaffected by ambiguous language, it is not without failures and limitations.

In Fig. 16, we visualize the failure modes of RT-Sketch. One failure mode we see with RT-Sketch  is occasionally re-trying excessively, as a result of trying to align the scene as closely as possible. For instance, in the top row, Rollout Image 3, the scene is already well-aligned, but RT-Sketch keeps shifting the chip bag which causes some misalignment in terms of the chip bag orientation. Still, this kind of failure is most common with RT-Goal-Image (Table 1), and is not nearly as frequent for RT-Sketch. We posit that this could be due to the fact that sketches enable high-level spatial reasoning without over-attending to pixel-level details.

One consequence of spatial reasoning at such a high level, though, is an occasional lack of precision. This is noticeable when RT-Sketch orients items incorrectly (second row) or positions them slightly off, possibly disturbing other items in the scene (third row). This may be due to the fact that sketches are inherently imperfect, which makes it difficult to reason with such high precision.

Finally, we see that RT-Sketch occasionally manipulates the wrong object (rows 4 and 5). Interestingly, we see that a fairly frequent pattern of behavior is to manipulate the wrong object (orange in row 4) to the right target location (near green can in row 4). This may be due to the fact that the sketch-generating GAN has occasionally hallucinated artifacts or geometric details missing from the actual objects. Having been trained on some examples like these, RT-Sketch can mistakenly perceive the wrong object to be aligned with an object drawn in the sketch. However, the sketch still indicates the relative desired spatial positioning of objects in the scene, so in this case RT-Sketch still attempts to align the incorrect object with the proper place.

Finally, the least frequent failure mode is manipulating the wrong object to the wrong target location (i.e. opening the wrong drawer handle). This is most frequent when the input is a free-hand sketch, and could be mitigated by increasing sketch detail (Table 2).

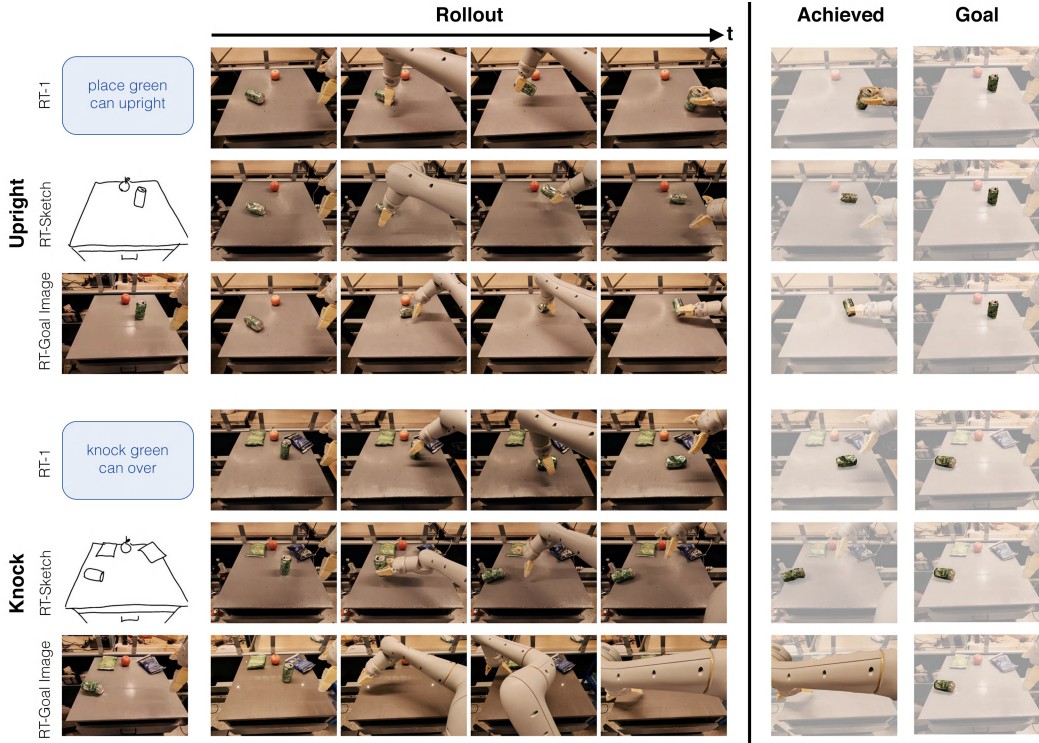

Figure 10: **H1 Rollout Visualization**: We visualize the performance of RT-1, RT-Sketch, and RT-Goal-Image on two skills from the RT-1 benchmark (*upright* and *knock*). For each skill, we visualize the goal provided as input to each policy, along with the policy rollout. We see that for both skills, RT-1 obeys the semantic task at hand by successfully placing the can upright or sideways, as intended. Meanwhile, RT-Sketch and RT-Goal-Image struggle with orienting the can upright, but successfuly knock it sideways. Interestingly, both RT-Sketch and RT-Goal-Image are able to place the can in the desired location (disregarding can orientation) whereas RT-1 does not pay attention to where in the scene the can should be placed. This is indicated by the discrepancy in position of the can in the achieved versus goal images on the right. This trend best explains the anomalous performance of RT-Sketch and RT-Goal-Image in perceived Likert ratings for the upright task (Fig. 3), but validates their comparably higher spatial precision compared to RT-1 across all benchmark skills (Table 1).

# G Evaluation and Assessment Interfaces

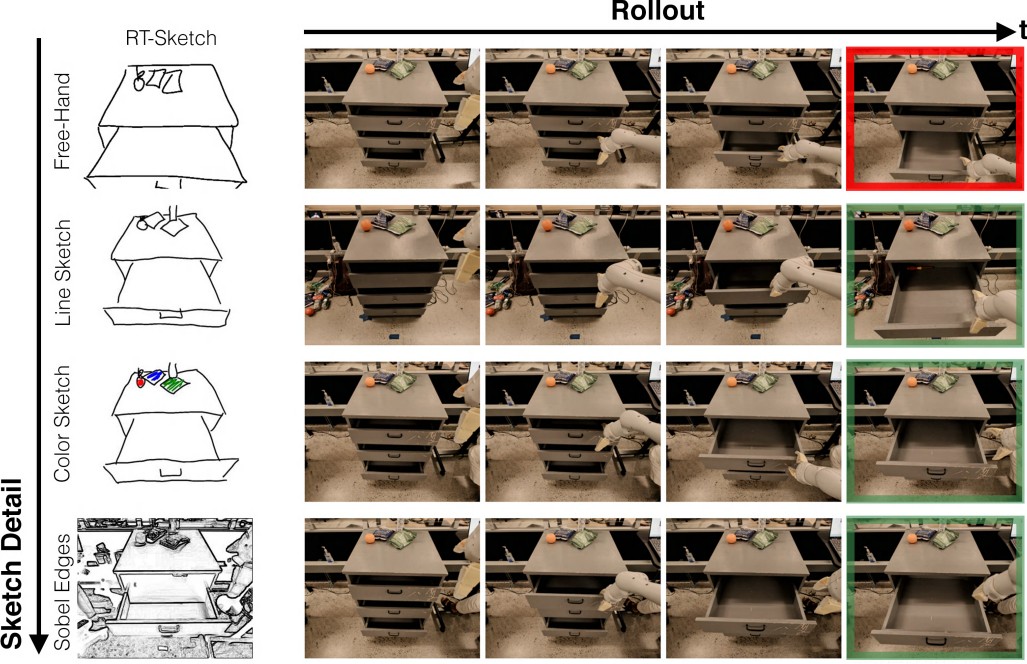

Figure 11: **H2 Rollout Visualization**: For the *open drawer* skill, we visualize four separate rollouts of RT-Sketch operating from different input types. Free-hand sketches are drawn without outlining over the original image, such that they can contain marked perspective differences, partially obscured objects (drawer handle), and roughly drawn object outlines. Line sketches are drawn on top of the original image using the sketching interface we present in Appendix Fig. 17. Color sketches merely add color infills to the previous modality, and Sobel Edges represent an upper bound in terms of unrealistic sketch detail. We see that RT-Sketch is able to successfully open the correct drawer for any sketch input except the free-hand sketch, without a noticeable performance gain or drop. For the free-hand sketch, RT-Sketch still recognizes the need for opening a drawer, but the differences in sketch perspective and scale can occasionally cause the policy to attend to the wrong drawer, as depicted.

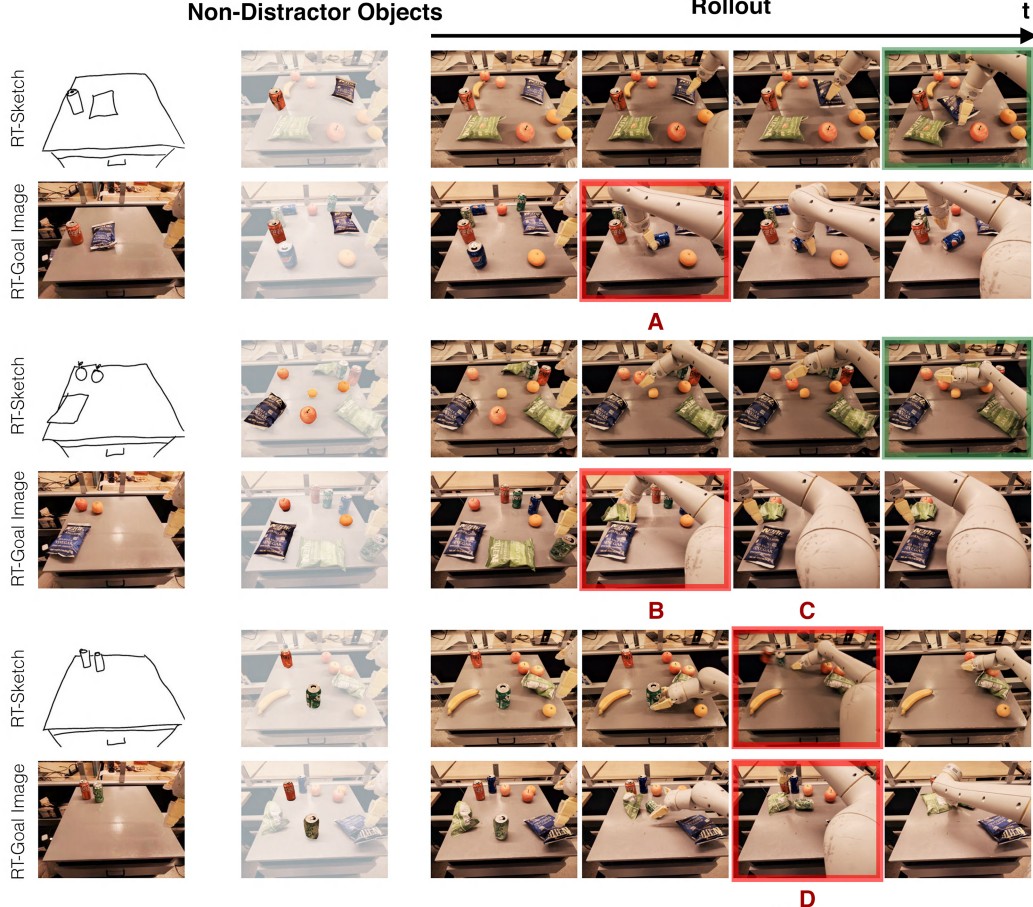

Figure 12: **H3 Rollout Visualization**: We visualize qualitative rollouts for RT-Sketch and RT-Goal-Image for 3 separate trials of the *move near* skill subject to distractor objects. In Column 2, we highlight the relevant non-distractor objects that the policy must manipulate in order to achieve the given goal. In Trial 1, we see that RT-Sketch successfuly attends to the relevant objects and moves the blue chip bag near the coke can. Meanwhile, RT-Goal-Image is confused about which blue object to manipulate, and picks up the blue pepsi can instead of the blue chip bag (A). In Trial 2, RT-Sketch successfully moves an apple near the fruit on the left. A benefit of sketches is their ability to capture instance multimodality, as any of the fruits highlighted in Column 2 are valid options to move, whereas this does not hold for an overspecified goal image. RT-Goal-Image erroneously picks up the green chip bag (B) instead of a fruit. Finally, Trial 3 shows a failure for both policies. While RT-Sketch successfully infers that the green can must be moved near the red one, it accidentally knocks over the red can (C) in the process. Meanwhile, RT-Goal-Image prematurely drops the green can and instead tries to pick the green chip bag (D).

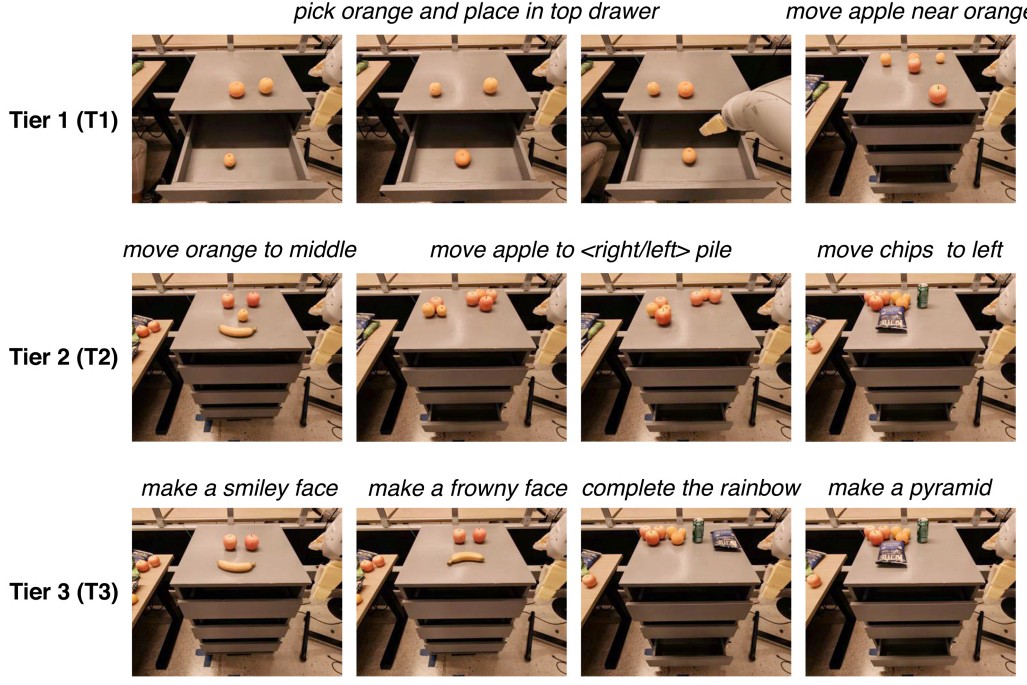

Figure 13: **H4 Tiers of Difficulty**: To test **H4**, we consider language instructions that are either ambiguous due the presence of multiple similar object instances (**T1**), are somewhat out-of-distribution for RT-1 (**T2**), or are far out-of-distribution and difficult to specify concretely without lengthier descriptions (**T3**). Each image represents the ground truth goal image paired with the task description.

**Rollout**

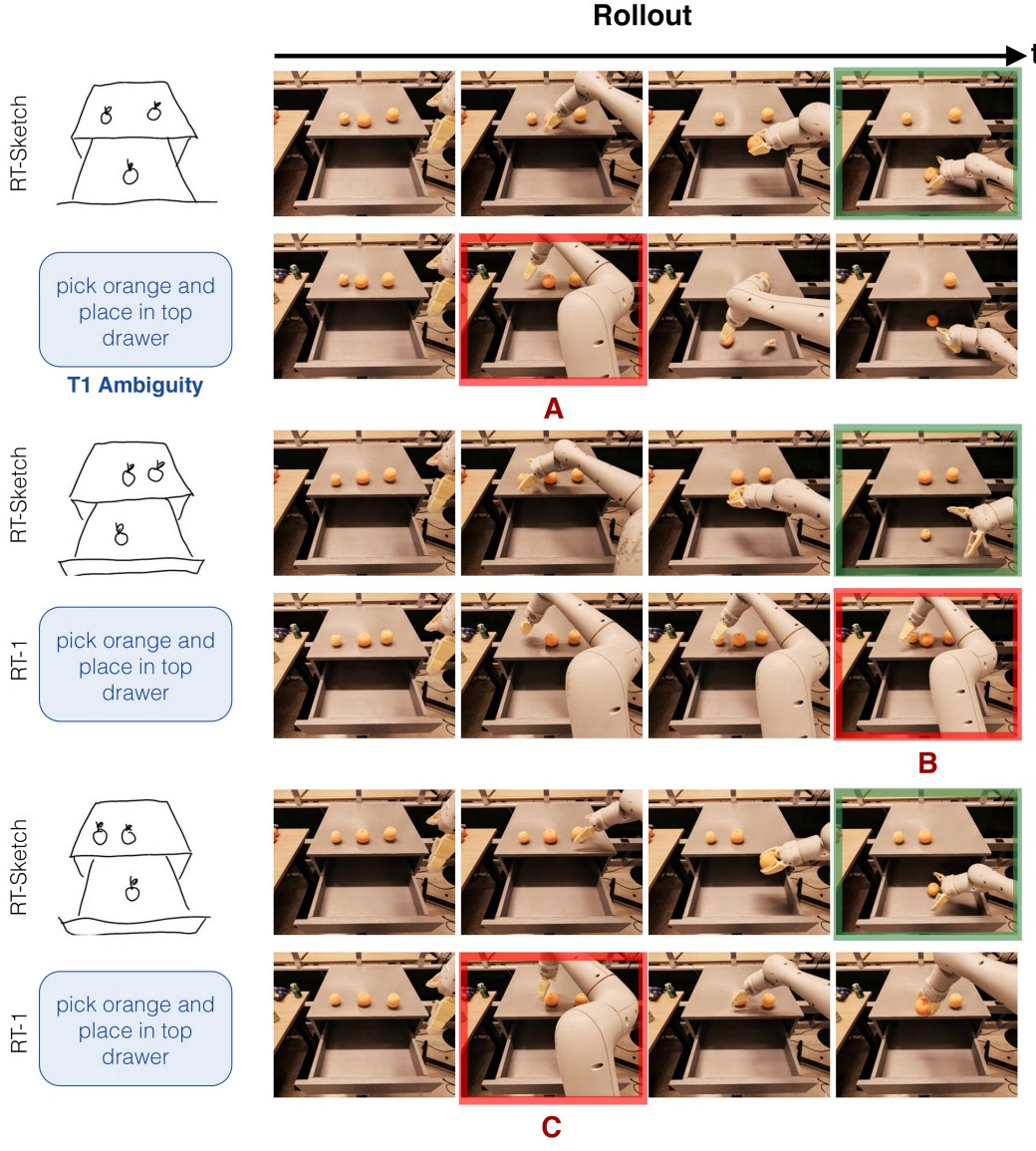

Figure 14: **H4 Rollout Visualization (T1 as visualized in Fig. 13)**: One source of ambiguity in language descriptions is mentioning an object for which there are multiple instances present. For example, we can easily illustrate three different desired placements of an orange in the drawer via a sketch, but an ambiguous instruction cannot easily specify which orange is relevant to pick and place. In all rollouts, RT-Sketch successfully places the correct orange in the drawer, while RT-1 either picks up the wrong object (A), fails to move to the place location (B), or knocks off one of the oranges (C). Although in this case, the correct orange to manipulate could easily be specified with a spatial relation like *pick up the ⟨ left/middle/right ⟩ orange*, we show below in Appendix Fig. 15 that this type of language is still out of the realm of RT-1's semantic familiarity.

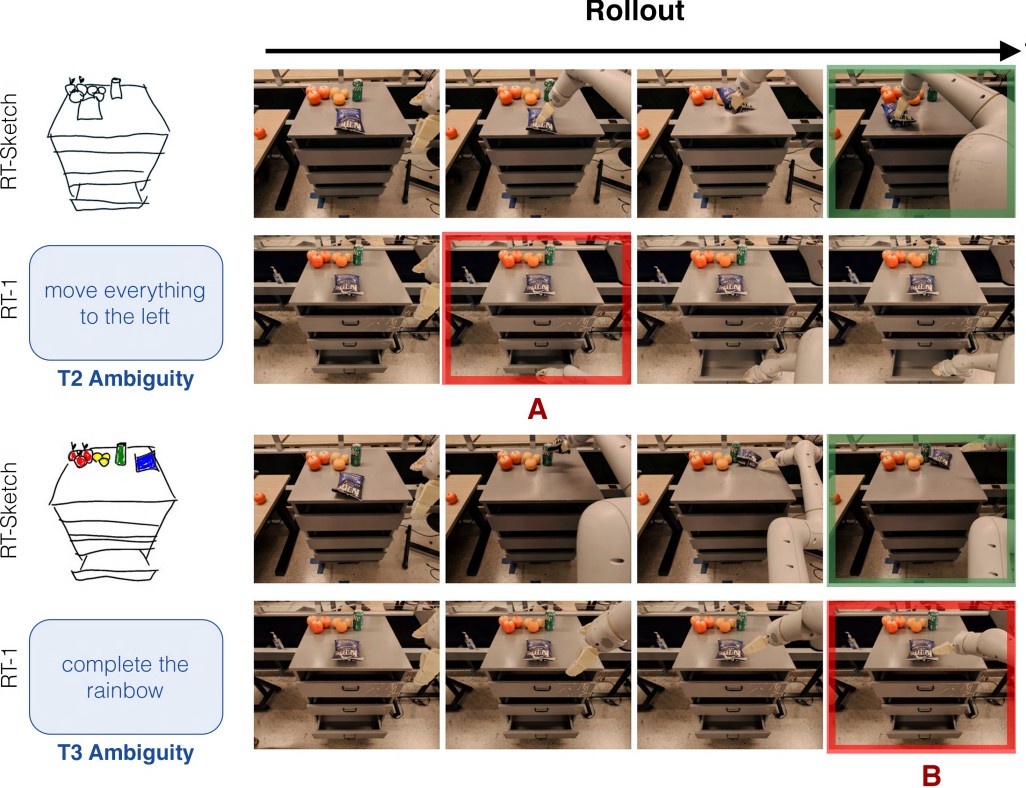

Figure 15: **H4 Rollout Visualization (T2-3 as visualized in Fig. 13)**: For **T2**, we consider language with spatial cues that intuitively should help the policy disambiguate in scenarios like the oranges in Fig. 14. However, we find that RT-1 is not trained to handle such spatial references, and this kind of language causes a large distribution shift leading to unwanted behavior. Thus, for the top rollout of trying to move the chip bag to the left where there is an existing pile, RT-Sketch completes the skill without issues, but RT-1 attempts to open the drawer instead of even attempting to rearrange anything on the countertop (A). For **T3**, we consider language goals that are even more abstract in interpretation, without explicit objects mentioned or spatial cues. Here, sketches are advantageous in their ability to succinctly communicate goals (i.e. visual representation of a rainbow), whereas the corresponding language task string is far too underspecified and OOD for the policy to handle (B).

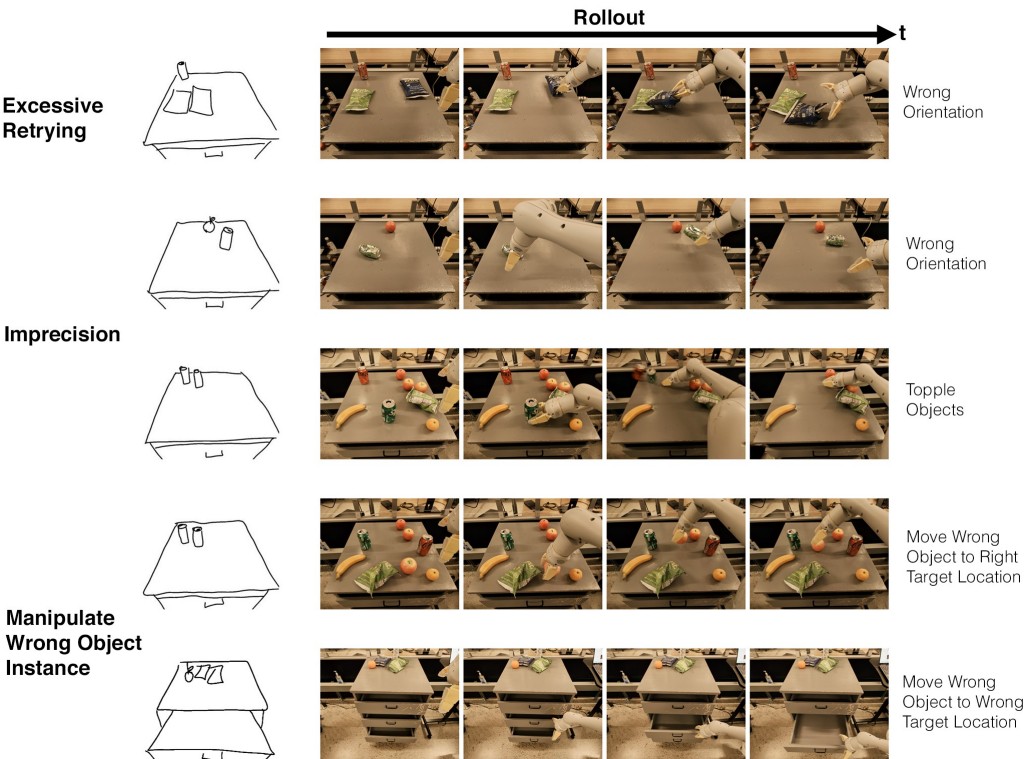

Figure 16: **RT-Sketch Failure Modes**

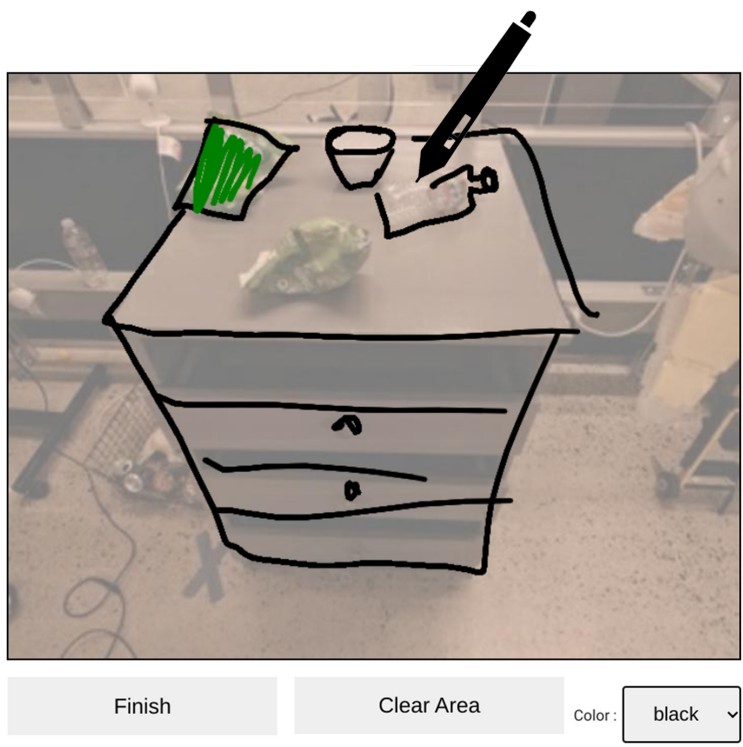

Figure 17: **Sketching UI**: We design a custom sketching interface for manually collecting paired robot images and sketches with which to train $\mathcal{T}$, and for sketching goals for evaluation. The interface visualizes the current robot observation, and provides the ability to draw on a digital screen with a stylus. The above visualization shows the color-sketching modality, which is a traced representation with color shading. The interface supports different colors and erasure, along with either *tracing* over the image (line-sketching) or drawing free-form over a blank canvas (free-hand sketches). We note that intuitively, drawing on top of the image is not an unreasonable assumption to make, since current agent observations are typically readily available compared to a goal image, for instance. Additionally, the overlay is intended to make the sketching interface easy for the user to provide, without having to eyeball edges for the drawers or handles blindly. This provides helpful guides for sketching and is an easy way to obtain sketches that more closely align with current observations for free.

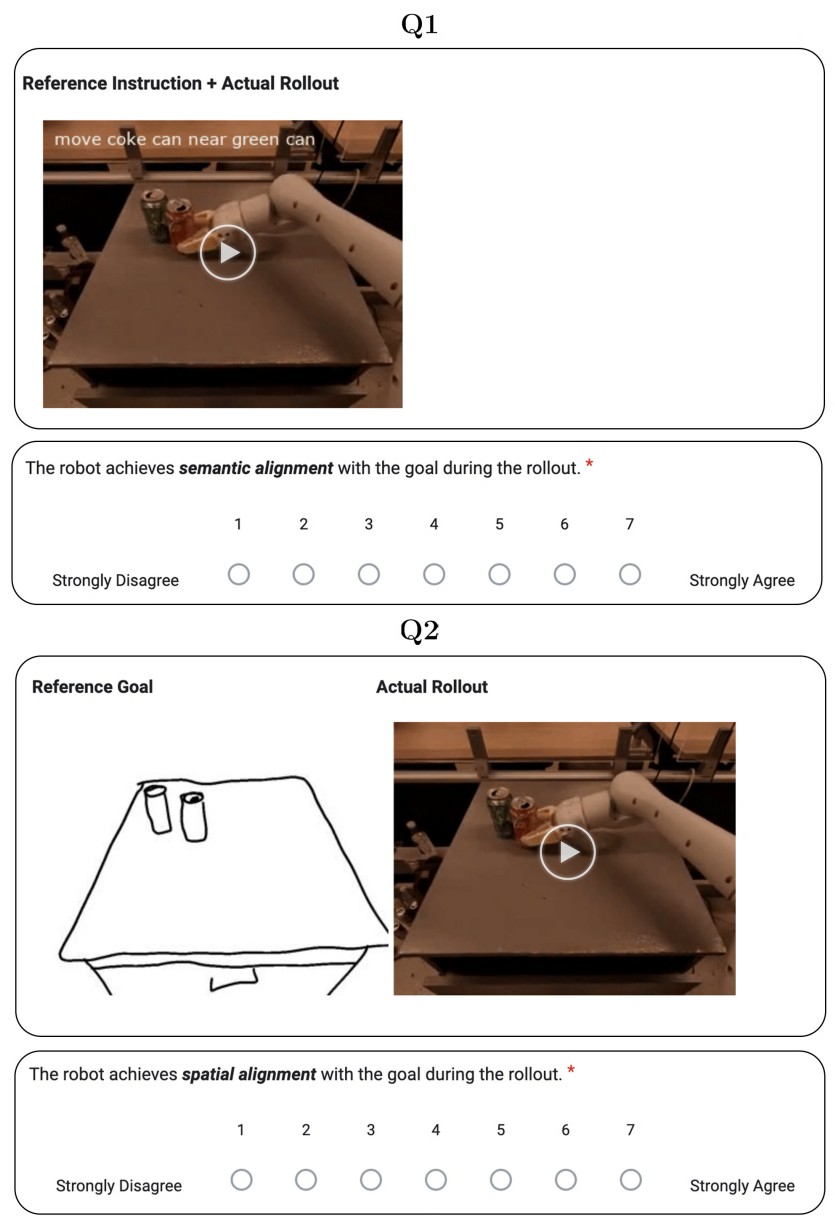

Figure 18: **Assessment UI**: For all skills and methods, we ask labelers to assess semantic and spatial alignment of the recorded rollout relative to the ground truth semantic instruction and visual goal. We show the interface above, where labelers are randomly assigned to skills and methods (anonymized). The results of these surveys are reported in Fig. 3.

