# OpenReview forum: "RT-Sketch: Goal-Conditioned Imitation Learning from Hand-Drawn Sketches"
_robot-learning.org/CoRL/2024/Conference — CoRL 2024_

### Official Review · Reviewer_SdeT · 2024-07-20
**A creative task definition with a successful method.**

**Originality:** 5
**Technical Quality:** 5
**Clarity Of Presentation:** 5
**Potential Impact:** 3
**Recommendation:** 4
**Confidence:** 4

**Review:**

This paper was of high quality. It was very clearly written and organized. It was very creative and original for the authors to think of sketches as a modality to control robots.  This work is highly significant since it introduces a new input to robots and compares it to other inputs.

There are several strengths of the paper. The writing and organization of the paper are very clear. The authors do thorough evaluation of the presented model. The four tested hypotheses offer a thorough investigation of what might be useful or advantageous of using sketches as inputs to robots. The authors tie their work well to its motivation. The video summarized the paper very clearly.

There are no huge weaknesses of this paper that I could determine. The authors acknowledge that the tasks are relatively limited to the training tasks. It would be nice to assess the user experience with controlling a robot via sketches. Sketches can be difficult and time consuming compared to language instructions. I would like to see a user study comparing the experience of using different modalities, but I think that that is out of scope for this paper.

**Quality Of The Limitations Section:**

3

**Questions For Rebuttal:**

The participants in the evaluation were unpaid. What was the general motivation for these people to participate?

The table setting example is compelling. I agree it would be very challenging to describe these arrangements with language. Could you please have the robot perform a demonstration of this example? If it is infeasible to try this, could you please explain whether you think RT-Sketch would be capable of doing this?

MISC:
Typo Line 321: "RT-Sketchachieves"

**Robotics Focus:**

4

**Summary Of Paper:**

This paper introduces the task of using sketches as a way to command a robot to do table top and cabinet related tasks. The compare this to language and photograph conditioned methodologies. They show that sketches, as input have a great advantage over other inputs in a variety of circumstances.

**Summary Of Recommendation:**

This paper proposes an interesting novel task. The paper is well written and the evaluation is thorough with interesting results.

---

### Official Review · Reviewer_LJYq · 2024-07-22
**Well-presented and novel approach for goal-conditioning**

**Originality:** 4
**Technical Quality:** 3
**Clarity Of Presentation:** 5
**Potential Impact:** 4
**Recommendation:** 3
**Confidence:** 4

**Review:**

Strengths:

I find the contributions of this paper simple yet well-motivated. Sketches can help disambiguate task-relevant from distractor objects, and are easy to provide on-the-fly just like language. Such kind of exploration for goal-conditioned policy learning is impactful for the field of robotics. I would also categorize this paper to be one that tries finding ways of using non-robotics datasets (contour drawing dataset in this case) to help improve visuomotor learning performance, which is an interesting and potentially useful direction in robot learning.

Weaknesses:
1. The suite of tasks considered for evaluating the policy could be diversified. Many tasks considered in recent papers are object-centric and have low tolerances for error (e.g. Stanford’s “make coffee” task) which creates bottlenecks that drastically reduces task success rates. Can the authors consider experimenting on such high tolerance tasks, or add discussion in limitations as to which kinds of tasks would have to be excluded for now given the reported mean-squared errors?
2. Why did the authors not experiment with the state-of-the-art Diffusion Policy with sketch and real images as goals?

**Quality Of The Limitations Section:**

2

**Questions For Rebuttal:**

Please refer to weaknesses mentioned above.

**Robotics Focus:**

4

**Summary Of Paper:**

The paper presents a goal-conditioned policy for manipulation using “sketches” as the conditioning in contrast to language or camera images, which the authors claim can be ambiguous or contain distractors respectively. Results are quantitatively good, and can drive future multimodal goal-specification research.

**Summary Of Recommendation:**

I believe the paper presents a novel approach for goal-conditioning that can drive future multimodal goal-specification research. However, some more experiments/discussion might be needed to convince readers of potential general efficacy of this approach in robot learning.

---

### Official Review · Reviewer_C69L · 2024-07-24

**Originality:** 3
**Technical Quality:** 3
**Clarity Of Presentation:** 4
**Potential Impact:** 2
**Recommendation:** 3
**Confidence:** 4

**Review:**

This paper presents an approach for goal-conditioned imitation learning, where the goal is specified by a user sketch. To tackle this, the authors propose RT-sketch -- essentially a robot transformer model that additionally conditions on goal images. The technical contributions here mainly lie in the construction of the sketch dataset: this dataset contains "sketches" generated by edge detectors and generative networks, as well as real manually sketched goal images. These are then conditioned on to train robot policies with transformer models. A major focus of this work is to get diverse sketches in the dataset so that it can handle a variety of sketches. Various permutations, such as colorization of part os the sketch have been investigated.

The motivations here are justified -- sketches provide a fairly specific and precise interface to communicate with robots that would be challenging to achieve with language, especially when considering spatial attributes. The weaknesses are in the practicalities of the system, the sketches provided are either quite detailed and accurate even when the problem setting is quite simple. e.g. the chest of drawers sketched have handles on each of the individual drawers, even when most of these are irrelevant; the number of circular objects in the bowl matches the real number (the sketch has 8 objects in the bowl in fig 7.). It's reasonable to expect that the method breaks at some level of abstraction: I would be interested to see additional experiments to identify where this boundary of the level of abstraction is.

Finally, although sketching desired goals is novel, there have been attempts to integrated sketches into specifications for robots that are note referenced here. These include sketching to instruct motion trajectories [1] and sketching to specify affordances [2]. These are relevant related works that should be mentioned.

[1] Zhi et al., Instructing Robots by Sketching: Learning from demonstration via probabilistic diagrammatic teaching, 2024.
[2] Masnadi et al., A Sketch-Based System for Human-Guided Constrained Object Manipulation, 2020.

**Quality Of The Limitations Section:**

3

**Questions For Rebuttal:**

Please see review above.

**Robotics Focus:**

4

**Summary Of Paper:**

Please see review below

**Summary Of Recommendation:**

Please see review above.

---

### Author Rebuttal · Authors · 2024-08-12

Here, we provide the updated manuscript, with new changes highlighted in blue, as a new pdf (revised.pdf). The remaining rebuttal updates are primarily **new real-world evalutions** best viewed at [this link](https://rt-sketch-anon.github.io/rebuttal.html).

---

### Decision · Program_Chairs · 2024-09-04

**Decision:**

Accept

**Comment:**

The reviewers were positive about this paper. There are only a couple of questions that the authors might comment on:

-- At what level of abstraction is the sketch no longer able to convey goal effectively? It would be interesting to probe this question.

-- How might we generate a more diverse set of sketches? This seems related to the first question.

Post review: good job.